# M3D-skin: Design and Applications of Multi-material 3D-printed Tactile Sensors with Hierarchical Infill Structures

Shunnosuke Yoshimura[1], Kento Kawaharazuka[1,2], and Kei Okada[1]

*Abstract*— The applicability of tactile sensors can be significantly broadened if they can be fabricated easily in diverse shapes. To address this challenge, we propose M3D-skin, a tactile sensor that leverages the infill patterns of layered 3D printing as the sensing principle, enabling simple fabrication in arbitrary shapes. Our approach employs flexible conductive and non-conductive filaments to form hierarchical structures defined by specific infill patterns. These flexible hierarchical structures deform under pressure, resulting in resistance changes that allow the acquisition of tactile information. We evaluate the fundamental characteristics of the proposed sensor and further demonstrate applications by integrating it with a robotic hand to enable tactile-based actions. Through these experiments, we show the effectiveness of the proposed tactile sensor.

## I. INTRODUCTION

In recent years, 3D printing has become increasingly accessible for component fabrication. Flexible and functional materials can now be readily printed into components using fused deposition modeling (FDM) 3D printers. Recent advances in 3D-printed tactile sensors ranging from PolyJet-based hands [1] to piezoresistive and capacitive FDM sensors using conductive or elastomeric filaments [2]–[5] demonstrate the potential of low-cost and shape-customizable fabrication using common materials. In this study, we develop M3D-skin [6], a pressure sensor that can be easily fabricated and integrated by employing a multi-material 3D printer capable of printing these materials in a single process. An overview of the proposed sensor is shown in Fig. 1.

## II. METHOD

### A. Sensor Structure

This sensor is fabricated using a fused deposition modeling (FDM) 3D printer with conductive and non-conductive thermoplastic polyurethane (TPU) filaments. The sensing structure is shown in Fig. 2. It is formed by alternately stacking conductive and non-conductive filaments in a sparse infill pattern. In the 3D model at the design stage, the conductive filament layers are completely separated from each other.

However, during printing, overhangs and instabilities within the sparse infill pattern cause portions of the conductive filament to protrude into the adjacent non-conductive

[1] The authors are with the Department of Mechano-Informatics, Graduate School of Information Science and Technology, The University of Tokyo, 7-3-1 Hongo, Bunkyo-ku, Tokyo, 113-8656, Japan. [yoshimura, kawaharazuka, okada]@jsk.t.u-tokyo.ac.jp

[2] The author is with the AI Center, Graduate School of Information Science and Technology, The University of Tokyo, Japan.

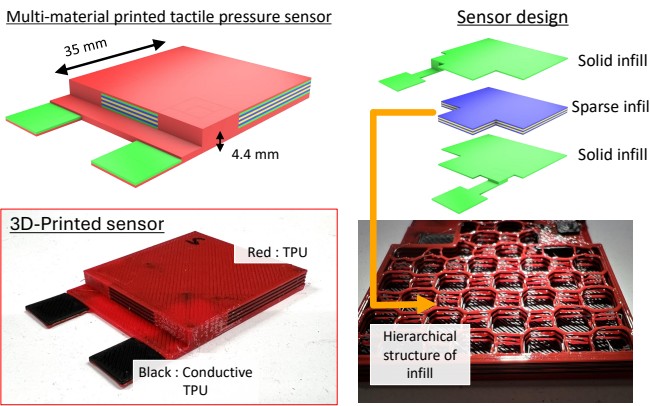

Fig. 1. Overview of this study, showing the 3D-printed sensor M3D-skin and its internal structure with infill.

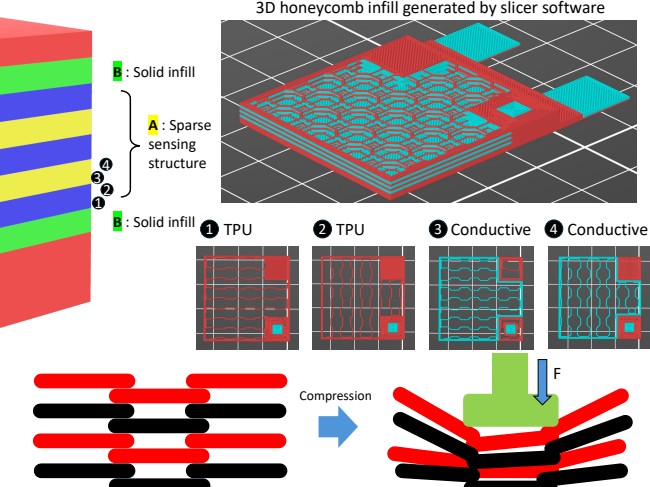

Fig. 2. The structure and principle of the sensing layer are illustrated. In particular, the sparse infill-based pattern structure is shown both in its overall configuration and in the shape of each individual layer.

filament layers. As a result, even in the undeformed state, the conductive filament layers may establish contact with the layers above and below, creating an initial conductive path. When pressure is applied, the sparse and flexible infill structure deforms, further increasing the contact area between conductive filament layers. This deformation leads to a change in electrical resistance, which enables pressure sensing.

The key design parameters are listed in Table I.

### B. Fabrication

For sensor fabrication, we employed the Prusa XL 5-toolhead, a tool-changer type multi-material FDM 3D printer.

## TABLE I
### Baseline Design Parameters

| Parameter | Value |
| --- | --- |
| Sensor Layer Conductive Filament Thickness | 0.4 mm |
| Sensor Layer Non-Conductive Filament Thickness | 0.4 mm |
| Sensor Layer Number of Patterned Layers | 4 |
| Wiring Layer Thickness | 0.4 mm |
| Cover Layer Thickness | 0.4 mm |
| Layer Height | 0.2 mm |
| Nozzle Diameter | 0.4 mm |
| Sensor Layer Infill Density | 10 % |
| Sensor Layer Infill Pattern | 3D Honeycomb |
| Sensor Layer Wall Thickness | 0.8 mm |
| Infill Density (Outside Sensor Area) | 100 % |

The non-conductive filament used for printing was TPU95A, while the conductive filament was TPU92A. The sensor was fabricated in a single process through monolithic printing, requiring no pre-processing, post-processing, or assembly.

## III. Experiment

### A. Sensor Characterization

First, we measured the characteristics of the sensor fabricated based on the shape shown in Fig. 1 and the parameters listed in Table I. A force was applied to the center of the sensor using a force gauge with a circular tip of 15 mm in diameter, and the change in resistance was measured with an Arduino via a voltage divider circuit. The results are shown in Fig. 3.

On the left side of the figure, the time variations of force and resistance are presented. In the undeformed state, the resistance was 5.9 kΩ. When a force was applied, the resistance initially increased to approximately 6.1 kΩ up to around 25 N, and then gradually decreased to 5.4 kΩ as the force increased to about 160 N. As the force was reduced, the resistance returned toward its initial value, reaching 6.1 kΩ when the force was released to 0 N. Although the resistance at 0 N was about 0.2 kΩ higher than the initial state, it gradually decreased over time and returned to its original value. This residual resistance change is considered to result from internal deformation remaining after the application of large forces, which slowly recovers over time.

On the right side of Fig. 3, the relationship between force/pressure and resistance is shown for the interval from $t = 3.5$ s to $t = 5.5$ s. Up to about 100 N (or 0.6 MPa), the resistance exhibited large changes, whereas beyond this range the change became smaller, showing only gradual variation up to 160 N. Relative to the initial resistance of approximately 6 kΩ, a change of about 0.6 kΩ was observed, which is sufficient to be reliably detected using the Arduino and voltage divider circuit.

### B. Sensor Applications

The application of the M3D-skin to a robotic hand is shown in Fig. 4. Further applications to different shapes and functions, including body measurement, are detailed in [6]. In this experiment, a four-tile sensor was attached to the right hand of the dual-arm robot PR2 to perform object

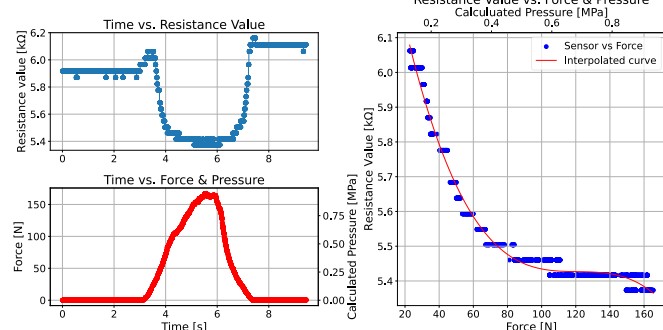

Fig. 3. Change in resistance when the sensor is subjected to an external force.

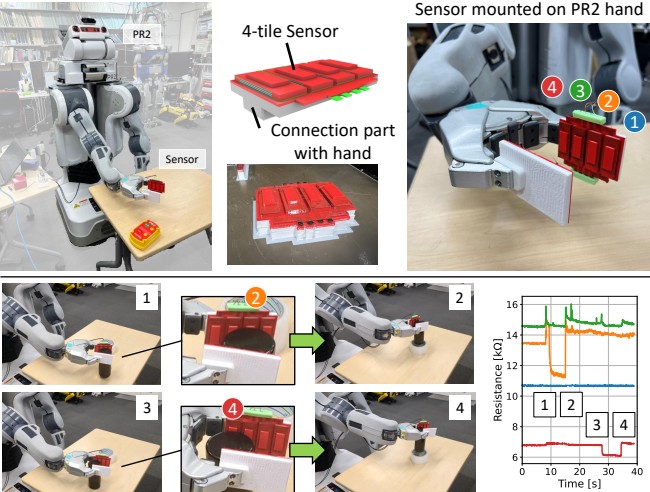

Fig. 4. Integration into a robotic hand and recognition task.

grasping and recognition tasks. The sensor was fabricated by monolithic printing with PLA filament integrated into the base for connection to the hand, and protrusions were added on the sensor surface to enhance sensitivity.

Using this hand, the change in sensor values during grasping was used to determine whether the object was positioned toward the fingertip side or the proximal side of the hand, enabling adjustment of the hand position to insert the object into a cylinder. In the first trial, the change in sensor 2 indicated that the object was located on the fingertip side, while in the second trial, the change in sensor 4 indicated that the object was on the proximal side.

This experiment demonstrated not only the applicability of the proposed sensor to robotic systems but also the ease of sensor integration achieved through monolithic fabrication with a 3D printer.

## IV. Conclusion

In this study, we proposed a tactile sensor that leverages the infill patterns of fused deposition modeling (FDM) 3D printing. Through experiments, we demonstrated that the proposed method provides a simple and versatile approach to tactile sensing, enabling fabrication in arbitrary shapes and suitability for embedded applications.

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
