# OpenReview forum: "M3D-skin: Design and Applications of Multi-material 3D-printed Tactile Sensors with Hierarchical Infill Structures"
_IEEE.org/IROS/2025/Workshop/Tactile_Sensing — IROS 2025 Workshop Tactile Sensing Poster_

### Official Review · Reviewer_N5V6 · 2025-09-13
**Some suggestions for M3D-skin**

**Rating:** 6
**Confidence:** 3

**Review:**

This paper proposes a novel processing method for fabricating tactile sensors. The overall logic of the paper is relatively clear, but the following shortcomings could be addressed and improved
1.	The author said in the manuscript: The applicability of tactile sensors can be significantly broadened if they can be fabricated easily in diverse shapes. However, this feature is not reflected in the article.
2.	It is recommended to add thickness information in Figure 1 to more intuitively show the volume of the sensor.
3.	In what ways does the sensor performance provided by this manuscript offer advantages over similar sensors?
4.	It is recommended to give specific parameters such as resolution accuracy, perception range, etc.

---

### Official Review · Reviewer_V9q1 · 2025-09-15
**Good paper, but needs more clarity in certain areas.**

**Rating:** 7
**Confidence:** 4

**Review:**

The paper is well written and promotes a tactile sensor design that can be 3D printed across various shapes and sizes. Good experimentation has been done to show material properties and drawbacks such as the residual difference from the deformation. Clear parameters of the experiments are outlined. To make the paper stand out more, or indeed be publishable beyond the workshop, the author will need to address the following issues:

1. The author claims their design can be used across different shapes, but has not shown this.

2. The author should give more thoughts on material deformation and how to combat this.

3. A more comprehensive literature review beyond self citation. There are various 3D printable tactile sensors out there, the author should consider the 3D printed magnetic skins (https://arxiv.org/abs/2506.09994) as a comparison. The authors having the benefit of size and potentially less noise than the mentioned paper.

4. The authors citation should include the place of publication in the existing citation (put "in press" or "submitted" or a DOI to an archived publication if awaiting publication).